# Patient-specific iPSC-derived photoreceptor precursor cells as a means to investigate retinitis pigmentosa

**Budd A Tucker[1], Robert F Mullins[1], Luan M Streb[1], Kristin Anfinson[1], Mari E Eyestone[1], Emily Kaalberg[1], Megan J Riker[1], Arlene V Drack[1], Terry A Braun[1,2], Edwin M Stone[1,3]\***

[1]Department of Ophthalmology and Visual Sciences, University of Iowa Carver College of Medicine, Iowa City, United States; [2]Department of Biomedical Engineering, University of Iowa Carver College of Medicine, Iowa City, United States; [3]Howard Hughes Medical Institute, University of Iowa, Iowa City, United States

**Abstract** Next-generation and Sanger sequencing were combined to identify disease-causing *USH2A* mutations in an adult patient with autosomal recessive RP. Induced pluripotent stem cells (iPSCs), generated from the patient's keratinocytes, were differentiated into multi-layer eyecup-like structures with features of human retinal precursor cells. The inner layer of the eyecups contained photoreceptor precursor cells that expressed photoreceptor markers and exhibited axonemes and basal bodies characteristic of outer segments. Analysis of the *USH2A* transcripts of these cells revealed that one of the patient's mutations causes exonification of intron 40, a translation frameshift and a premature stop codon. Western blotting revealed upregulation of GRP78 and GRP94, suggesting that the patient's other USH2A variant (Arg4192His) causes disease through protein misfolding and ER stress. Transplantation into 4-day-old immunodeficient *Crb1*[−/−] mice resulted in the formation of morphologically and immunohistochemically recognizable photoreceptor cells, suggesting that the mutations in this patient act via post-developmental photoreceptor degeneration.

**\*For correspondence:** edwin-stone@uiowa.edu

**Reviewing editor**: Jeremy Nathans, Johns Hopkins University School of Medicine, United States

## Introduction

Usher syndrome is a genetically heterogeneous autosomal recessive disorder characterized by early onset sensorineural hearing loss and later onset retinitis pigmentosa (RP). Mutations in the *USH2A* gene are the most common cause of Usher syndrome type I (*Aller et al., 2006*; *Baux et al., 2007*; *DePristo et al., 2011*) and are also a common cause of non-syndromic RP (*McGee et al., 2010*; *Vaché et al., 2012*). The combination of hearing loss and retinitis pigmentosa in Usher syndrome creates an unusual opportunity for the development of effective gene replacement therapy. Unlike many other forms of retinitis pigmentosa in which a large fraction of the photoreceptors have already been lost by the time a diagnosis is made, newborn hearing tests coupled with increasingly sensitive molecular testing have the potential to identify patients affected with Usher syndrome early enough that the majority of their photoreceptors are still amenable to gene replacement therapy.

The obstacles to such treatment include the large size of the *USH2A* gene, which precludes the use of the types of viral vectors currently employed for retinal gene therapy. Large genes also frequently harbor a number of rare variants of uncertain pathogenicity in the population, and these can make it difficult to establish a molecular diagnosis with sufficient certainty to undertake a therapy as invasive as subretinal injection of therapeutic viruses. Another obstacle to treatment is the relative paucity of information about the normal function of the protein encoded by *USH2A* (usherin) and the degree to which it can be overexpressed in human cells without causing harm.

**eLife digest** Retinitis pigmentosa is an inherited disorder in which the gradual degeneration of light-sensitive cells in the outer retina, known as photoreceptors, causes a progressive loss of sight. Retinitis pigmentosa can also occur as part of a wider syndrome: patients with Usher syndrome, for example, suffer from early-onset deafness and then develop retinitis pigmentosa later in life. Usher syndrome is caused by mutations in any of more than ten genes, but the most commonly affected is *USH2A*, which encodes a protein called usherin. Mutations in *USH2A* can also cause retinitis pigmentosa on its own.

Clinical trials are underway to determine whether it is possible to treat various forms of inherited retinal degeneration using gene therapy. This involves inserting a functional copy of the gene associated with the disease into an inactivated virus, which is then injected into the eye. The virus carries the target gene to the light-sensitive photoreceptor cells where it can replace the faulty gene. This could be particularly useful for conditions such as Usher syndrome, in which the early-onset deafness makes it possible to diagnose retinitis pigmentosa before substantial numbers of photoreceptor cells have been lost.

For gene therapy to become a widely used strategy for the treatment of retinal degenerative disease, identification and functional interrogation of the disease-causing gene/mutations will be critical. This is especially true for large highly polymorphic genes such as *USH2A* that often have mutations that are difficult to identify by standard sequencing techniques. Likewise, viruses that can carry large amounts of genetic material, or endogenous genome editing approaches, will need to be developed and validated in an efficient patient-specific model system.

Tucker et al. might have found a way to address these problems. In their study, they used skin cells from a retinitis pigmentosa patient with mutations in *USH2A* to produce induced pluripotent stem cells. These are cells that can be made to develop into a wide variety of mature cell types, depending on the exact conditions in which they are cultured. Tucker et al. used these stem cells to generate photoreceptor precursor cells, which they transplanted into the retinas of immune-suppressed mice. The cells developed into normal-looking photoreceptor cells that expressed photoreceptor-specific proteins.

These results have several implications. First, they support the idea that stem cell-derived retinal photoreceptor cells, generated from patients with unknown mutations, can be used to identify disease-causing genes and to interrogate disease pathophysiology. This will allow for a more rapid development of gene therapy strategies. Second, they demonstrate that *USH2A* mutations cause retinitis pigmentosa by affecting photoreceptors later in life rather than by altering their development. This suggests that it should, via early intervention, be possible to treat retinitis pigmentosa in adult patients with this form of the disease. Third, the technique could be used to generate animal models in which to study the effects of specific disease-causing mutations on cellular development and function. Finally, this study suggests that skin cells from adults with retinitis pigmentosa could be used to generate immunologically matched photoreceptor cells that can be transplanted back into the same patients to restore their sight. Many questions remain to be answered before this technique can be moved into clinical trials but, in the meantime, it will provide a new tool for research into this major cause of blindness.

The advent of induced pluripotent stem cells (iPSCs) (**Takahashi and Yamanaka, 2006**) and the ability to make tissue-specific progenitors from these cells have created a path for overcoming many of these obstacles. It is now possible to investigate the function and dysfunction of a disease-associated gene in tissues such as retina that are inaccessible to molecular analysis in living patients (**Tucker et al., 2011**; **Jin et al., 2012**; **Singh et al., 2013**). For instance, in a recent study, we sequenced the exome of an individual with RP who had no family history of eye disease and only one living sibling and identified a likely disease-causing homozygous mutation in a gene (MAK) that had not been previously reported to be associated with disease (**Tucker et al., 2011**). We validated this finding in a large cohort of RP patients and then used fibroblast-derived iPSCs from the proband to investigate the mechanism through which the mutation causes disease (**Tucker et al., 2011**).

The generation of iPSCs from older individuals is more difficult and less efficient than the generation of iPSCs from young individuals (**Mahmoudi and Brunet, 2012**). To help overcome this limitation, a

variety of different reprogramming factors, reprogramming enhancers, and cell types have been evaluated (*Liao et al., 2008*; *Judson et al., 2009*; *Mali et al., 2010*; *Cheng et al., 2011*; *Niibe et al., 2011*; *Szablowska-Gadomska et al., 2011*; *Zhang et al., 2011*; *Li and Rana, 2012*; *Lin et al., 2012*; *Liu et al., 2012*; *Mahmoudi and Brunet, 2012*; *Okita et al., 2013*; *Zhang and Wu, 2013*). Of the accessible cell types used for iPSC generation, the reprogramming of keratinocytes has been shown to be as much as 100-fold more efficient and at least twofold faster than the reprogramming of dermal fibroblasts (*Aasen et al., 2008*). In addition, it has recently been shown that keratinocyte-derived iPSCs are more similar to embryonic stem cells than those generated from fibroblasts (*Barrero et al., 2012*).

The ability to create otherwise inaccessible tissues like retina from patient-derived iPSCs provides a valuable tool for the study of disease pathophysiology and the development of treatment. However, the utility of this approach is limited by one's ability to generate relatively pure cultures of the cell type of interest. As clearly demonstrated in a variety of recent publications, early passage iPSCs tend to retain an epigenetic profile that is characteristic of their somatic tissue of origin (*Marchetto et al., 2009*; *Kim et al., 2011*). An excellent example of how epigenetic memory can influence retinal differentiation is the recent study by Clegg et al. (*Hu et al., 2010*), who showed that iPSCs generated from retinal pigmented epithelium (RPE) would preferentially re-differentiate into mature RPE. Although viable retinal tissue suitable for iPSC generation would be difficult to obtain for the purpose of patient-specific disease modeling and therapy, accessible somatic cells from the same germ lineage as the neural retina are readily available in the form of keratinocytes.

In addition to being a useful tool for investigation of disease pathophysiology, iPSC technology provides a means for future autologous photoreceptor transplantation for the treatment of retinal degeneration. Work from Ali et al. clearly demonstrates that the post-mitotic photoreceptor precursor cell is the optimal cell type for efficient rod photoreceptor cell replacement (*Lakowski et al., 2011*; *Pearson et al., 2012*). A variety of different protocols, utilizing both two- and three-dimensional culture systems, have succeeded in deriving photoreceptor precursor cells from less differentiated precursors (*Osakada et al., 2008*; *Hirami et al., 2009*; *Meyer et al., 2009*; *Osakada et al., 2009*; *Lamba et al., 2010*; *Meyer et al., 2011*; *Tucker et al., 2011*; *Nakano et al., 2012*; *Phillips et al., 2012*; *Sasai et al., 2012*; *Homma et al., 2013*; *Mekala et al., 2013*; *Tucker et al., 2013*). Although cultured three-dimensional eyecups will undoubtedly have many applications in developmental biology (*Nakano et al., 2012*; *Sasai et al., 2012*), two-dimensional systems have the advantage of easier identification and isolation of specific cell types for post-differentiation subculture. We and others have shown that following transplantation, iPSC-derived photoreceptor precursor cells give rise to rod and cone photoreceptor precursors, which integrate within the dystrophic retina, form synapses with host bipolar cells, and induce a partial restoration of electrophysiological and anatomical correlates of retinal function (*Lamba et al., 2010*; *Tucker et al., 2011*; *Homma et al., 2013*). Isolation of specific cell types from two-dimensional iPSC-derived eyecups followed by transplantation into animal eyes provides a means for exploring human retinal pathophysiology in unprecedented detail, as well as a means for efficiently evaluating gene-based and cell-based therapies.

In this study, we combined Sanger sequencing, next-generation sequencing, and iPSC technologies to identify the disease-causing *USH2A* mutations in a 62-year-old patient with retinitis pigmentosa and to demonstrate that the mutations we identified are present in the patient's retinal transcripts. In addition, we were able to show that the patient's iPSC-derived rod photoreceptor precursor cells could integrate within the immune suppressed developing mouse retina and give rise to morphologically and immunohistochemically recognizable photoreceptor cells. These findings suggest that the *USH2A* mutations in this patient act via post-developmental photoreceptor degeneration rather than an early developmental abnormality.

## Results

Genomic DNA from a patient affected with autosomal recessive retinitis pigmentosa (RP) was fragmented and hybridized to an Agilent exome capture reagent (v2), and the eluted fragments were sequenced on an Illumina sequencing instrument. This experiment yielded 71 million uniquely mapped paired-end sequences, each end 50 bp in length. These sequences were aligned to the reference human genomic sequence (hg19) using BWA (*Li and Durbin, 2009*) and BFAST (*Homer et al., 2009*). Departures from the reference were identified with GATK (*DePristo et al., 2011*). More than 20,000 sequence variations were detected (*Supplementary file 1B*). These variants were prioritized using the following criteria: GATK variation quality score greater than 50, presence in coding sequence or within 5 bp of a splice

junction, annotated frequency of less than 2% in all available population databases, presence in two or fewer exome sequences from other individuals analyzed by our laboratory, predicted to alter protein structure, and lack of complete genotype sharing with the proband's unaffected sibling (as determined by genomewide SNP genotyping) (*Supplementary file 1B*). After applying these filters to the data, more than 400 plausible disease-causing sequence variations, in more than 300 genes, remained for further consideration. There were no instances of two plausible disease-causing variants in a gene previously associated with RP, but two human retinal degeneration genes (*ABCA4* and *USH2A*) each harbored a single plausible disease-causing variant (Thr1428Met ACG>ATG and Arg4192His CGC>CAC, respectively). Sanger sequencing of the entire coding sequence of both of these genes confirmed the presence of the heterozygous variants identified by exome sequencing but failed to detect a second disease-causing variation in either gene. We then began evaluating genes that had not been previously associated with human retinal disease that contained two protein-altering variants in the proband. As we were screening additional RP patients and controls for these variants, two things occurred that re-directed our attention to *USH2A*. First, we discovered a second RP patient with the Arg4192His variant in *USH2A*, and this individual had a second well-established disease-causing mutation on the opposite *USH2A* allele. More importantly, *Vaché et al. (2012)* published their discovery of a disease-causing variant in intron 40 of *USH2A*. Sanger sequencing of intron 40 in our patient revealed this intronic variant to be present in *trans* to his Arg4192His mutation.

To validate the pathogenicity of these mutations in retinal tissue, and to begin developing a cell replacement therapy, we developed an iPSC line from our patient using a primary culture of keratinocytes as the starting material. Like retina, keratinocytes are of ectodermal origin and are significantly easier to reprogram than fibroblasts (*Aasen et al., 2008*). This is especially important when trying to generate iPSCs from older individuals.

The proband's keratinocytes (*Figure 1A*) were expanded for three passages prior to being reprogrammed into iPSCs (*Tucker et al., 2013*). For iPSC generation, passage-4 cells were plated at a density of 30,000 cells/cm$^2$ and transduced with the transcription factors OCT4, SOX2, KLF4, and C-MYC. Approximately, 7 days post-transduction, cells were passaged at a density of 10,000 cells/cm$^2$ onto fresh Synthemax cell culture plates and fed every other day with fresh iPSC induction media. 2–3 weeks following the initial passage, small, morphologically distinct cell clusters were present (*Figure 1B*). 2–3 weeks later, cell colonies large enough for mechanical isolation were dissected from the surrounding differentiated keratinocyte layer. Each isolated colony was dissociated into 150–200 μm square cell clusters and cultured in individual wells of a 24-well Synthemax cell culture plate. Each well was maintained as a separate clonally expanded line for four passages prior to analysis. At passage 4, cultures contained well-defined densely packed colonies consisting of cells with a high nucleus to cytoplasm ratio (*Figure 1C*, RP-iPSC). To test pluripotency, rt-PCR and teratoma assays were performed. rt-PCR analysis confirmed expression of the pluripotency markers DNMT, LIN28, OCT4, KLF4, SOX2, Nanog, and C-MYC (*Figure 1D*).

Histologic analysis of teratomas revealed tissues specific to each of the three embryonic germ layers (*Figure 1E,F*). Similarly, immunohistochemical staining revealed GFAP positive neural tissue (*Figure 1G*) and αSMA positive vascular structures (*Figure 1H*).

To derive retinal neurons from these iPSCs for pathophysiologic studies and the development of patient-specific therapy, a slightly modified version of our previously published stepwise differentiation protocol was used (*Figure 2A*) (*Tucker et al., 2011*, *2011*, *2013*). This protocol is designed to maximize the percentage of retinal cells produced by taking into account: (1) the role of bone morphogenic protein (BMP) and Wnt signaling pathway inhibition in neuroectodermal development (noggin and DKK1, respectively) (*Lamb et al., 1993*; *Mukhopadhyay et al., 2001*; *Anderson et al., 2002*); (2) the role of IGF-1 in anterior neural/eye field development (*Pera et al., 2001*); and (3) Notch pathway inhibition (DAPT—gamma secretase inhibitor) in photoreceptor cell development (*Jadhav et al., 2006*). Although retinal differentiation in mouse can be accomplished in 30 days (*Tucker et al., 2011*, *2013*), retinal differentiation of human cells takes significantly longer. During differentiation, these keratinocyte-derived iPSCs formed eyecup-like structures with clearly defined layers of pigmented RPE and non-pigmented neural retina (*Figure 2B–E*). This was a clear departure from our previous experience with fibroblast-derived iPSCs, which never develop eyecup-like structures under very similar culture conditions. The eyecups began as small pigmented foci that could be detected as early as 45 days post-differentiation (*Figure 2B*). Following passage and continued differentiation, these pigmented clumps expanded and elongated over 150 days (*Figure 2C*) and were eventually joined by

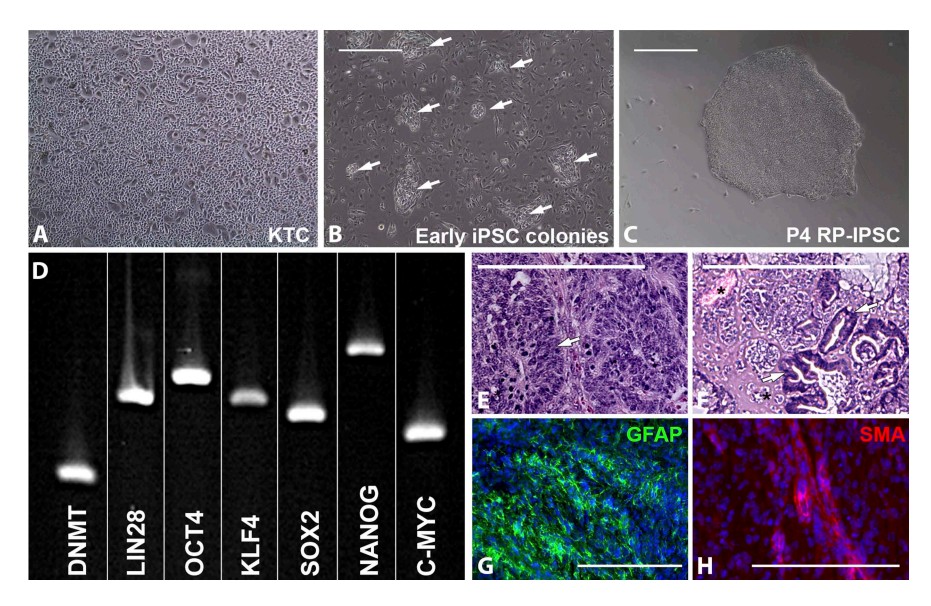

**Figure 1**. Derivation of iPSCs from keratinocytes of a patient affected with *USH2A*-associated RP. (**A**–**H**) Microscopic analysis of human keratinocytes (**A**), early keratinocyte-derived iPSC colonies (**B**, arrows), and purified keratinocyte-derived iPSC cultures (**C**). At 2–3 weeks post-viral transduction, ES-cell-like iPSC colonies begin to emerge (**B**, arrows). iPSC colonies isolated, subcultured, and expanded on Synthemax cell culture surfaces maintain a pluripotent morphology (**C**) express the pluripotency markers DNMT, LIN28, OCT4, KLF4, SOX2, NANOG, and C-MYC (**D**), and form teratomas consisting of tissues of ectoderm (**E**, arrow and **G**, GFAP in green), mesoderm (**F**, asterisk and **H**, SMA in red), and endoderm (**F**, arrows) each of the three embryonic germ layers (**E**–**H**). Scale bar = 400 µm.

a clump of neural rosettes (*Figure 2C*, arrow) on one side of the pigmented structure. In some cases, these RPE/neural units extended into C-shaped eyecup-like structures (*Figure 2D,E*) while in others, the central cavity did not form and the structure consisted of a circular sheet of neural cells surrounded by a pigmented epithelium (*Figure 2—figure supplement 1*). In many cases, multiple eyecups at different developmental stages were present within the same well of a differentiating six-well plate (*Figure 2D*, arrows). The reason that these eyecup-like structures develop in a planar fashion instead of the spheres reported by both Meyer et al. and Eiraku et al. is that we grow these cells under adherent conditions in tissue culture dishes coated with collagen, laminin, and fibronectin, while spheres are grown in suspension (*Meyer et al., 2011*; *Eiraku and Sasai, 2012*). Although there will undoubtedly be advantages of both approaches in different situations, two advantages of this planar system is that the neural cells in the center of the eyecup do not necrose for lack of oxygen, and the firm attachment to the underlying substrate allows homogeneous biopsies to be taken from the eyecups for subculture and analysis.

For example, to explore the cellular makeup of the pigmented layer of the eyecups (*Figure 3A*), biopsies were taken, gently dissociated, and subcultured for subsequent expansion and microscopic analysis. 24 hr after plating, pigment-containing cell clusters adhered to the culture surface and began to give rise to non-pigmented cells with a fibroblastic morphology (*Figure 3B*). By 72–96 hr post-plating, extensive cell spreading and a complete loss of pigmentation was noted (*Figure 3C*). By 2 weeks post-plating, confluent cultures of densely pigmented hexagonal epithelial cells were present (*Figure 3D*). Immunocytochemical analysis of these cultures revealed cells expressing the tight junction marker ZO1 (*Figure 3E*), the transcription factor PAX6 (*Figure 3F*), the RPE-specific channel bestrophin (*Figure 3—figure supplement 1A*) and the RPE visual cycle protein RPE65 (*Figure 3—figure supplement 1B*). To further investigate the identity of cells within the pigmented layer of patient-specific eyecups, TEM analysis was performed. As shown in *Figure 3*, cells within the pigmented layer contained pigment granules (**G**, asterisks), tight junctions with neighboring cells (**G** and **H**, arrows), and apical microvilli (**G** and **H**, arrowheads), all of which are characteristic of native RPE. Collectively, these results indicate that the pigmented layer of cells located at the perimeter of patient-specific eyecups is patient-specific RPE.

**Figure 2**. Differentiation of human *USH2A*-associated iPSCs into eyecup-like structures. (**A**) Schematic diagram illustrating the differentiation paradigm utilized to generate human eyecup-like structures. (**B**–**E**) Morphological analysis of *USH2A*-associated iPSC-derived eyecups. iPSC-derived eyecups form pigmented cell clumps (**B**) that extend and wrap in a C shape around newly formed neural rossettes (**C**). Following this protocol, a typical six-well cell

*Figure 2. Continued on next page*

*Figure 2. Continued*

culture dish will have two to four eyecups/well, each at slightly different stages of development (**D**, arrows, a low magnification image of a typical well of a six-well plate with developing eyecups). At 150 days post-differentiation, complete eyecups with clearly defined neural retina and RPE layers can be identified (**D**, top right arrow, and **E**). Scale bar, **B** and **C** = 200 μm, **D** = 400 μm.

The following figure supplements are available for figure 2:

**Figure supplement 1**. Additional examples of human *USH2A*-associated iPSC-derived eyecup-like structures.

To determine whether cells contained within non-pigmented neural rosettes (*Figure 4A*) had adopted a photoreceptor cell fate, patient-specific eyecups were fixed and analyzed with both TEM (*Figure 4B*) and confocal microscopy using antibodies to the rod photoreceptor markers recoverin and rhodopsin (*Figure 4C–E*, and *Figure 2—figure supplement 1D–F*), and recoverin the connecting cilium marker acetylated tubulin (*Figure 4F–H*). Densely packed neural rosettes contained polarized

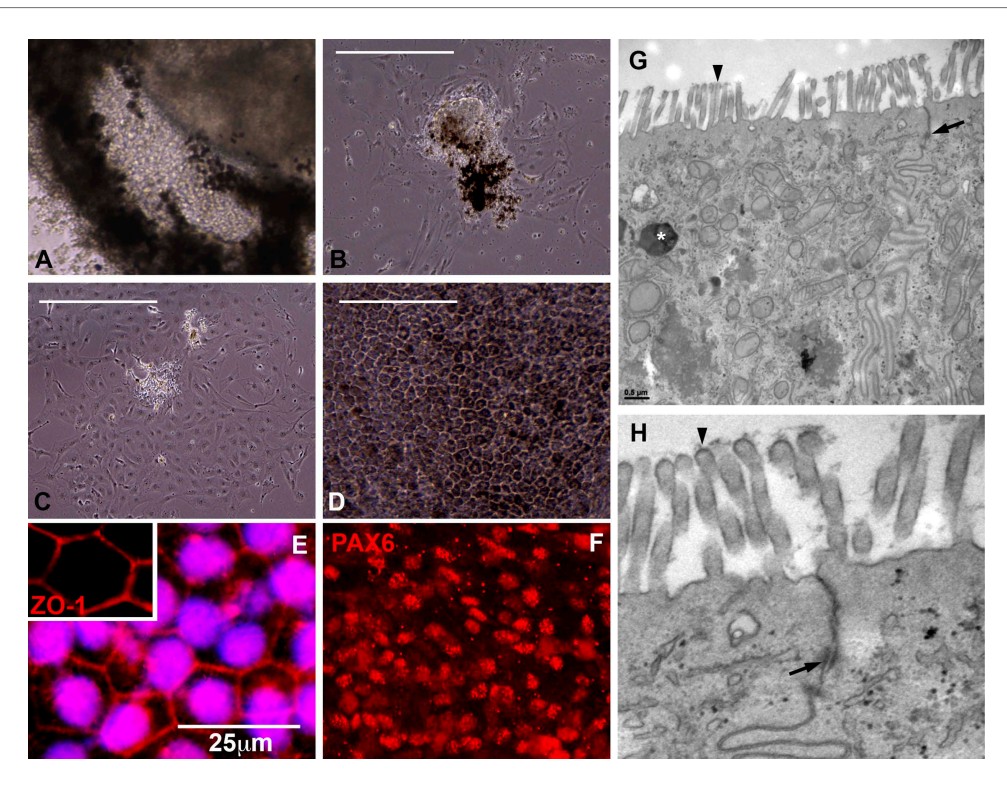

**Figure 3**. Cells contained within the pigmented layer of *USH2A*-associated eyecups are of RPE origin. (**A**) A high magnification phase image of the RPE layer of an *USH2A* eyecup prior to biopsy and subculture. (**B–D**) Area of the RPE presented in panel **A** was picked and subcultured in fresh RPE culture media on collagen, laminin, and fibronectin coated six-well culture dishes. 24 hr after plating, RPE cells spread and take on a fibroblastic morphology (**B**). By 72 hr post-plating, RPE cells lose their pigmentation and begin to form cell–cell contacts (**C**). 2 weeks post-plating, a confluent monolayer of RPE cells are present that have taken on the typical cuboidal RPE morphology and regained pigmentation (**D**). (**E–F**) Immunocytochemical analysis of *USH2A*-associated RPE cells with antibodies targeted against the tight junction marker ZO1 (**E**) and the transcription factor PAX6 (**F**). (**G–H**) TEM analysis of RPE cells within the intact RPE layer of *USH2A* eyecups (**H** is a high magnification view of the upper right corner of panel **G**). RPE cells are polarized, have apical microvilli, make tight junctions with neighboring RPE cells (**G** and **H**, arrows) and contain pigment granules within their cytoplasm (**G**, asterisk). Scale bar, **B–D** = 200 μm, **G** = 0.5 μm.

The following figure supplements are available for figure 3:

**Figure supplement 1**. Pigmented cells isolated from *USH2A*-associated eyecups express bestrophin 1 and RPE65.

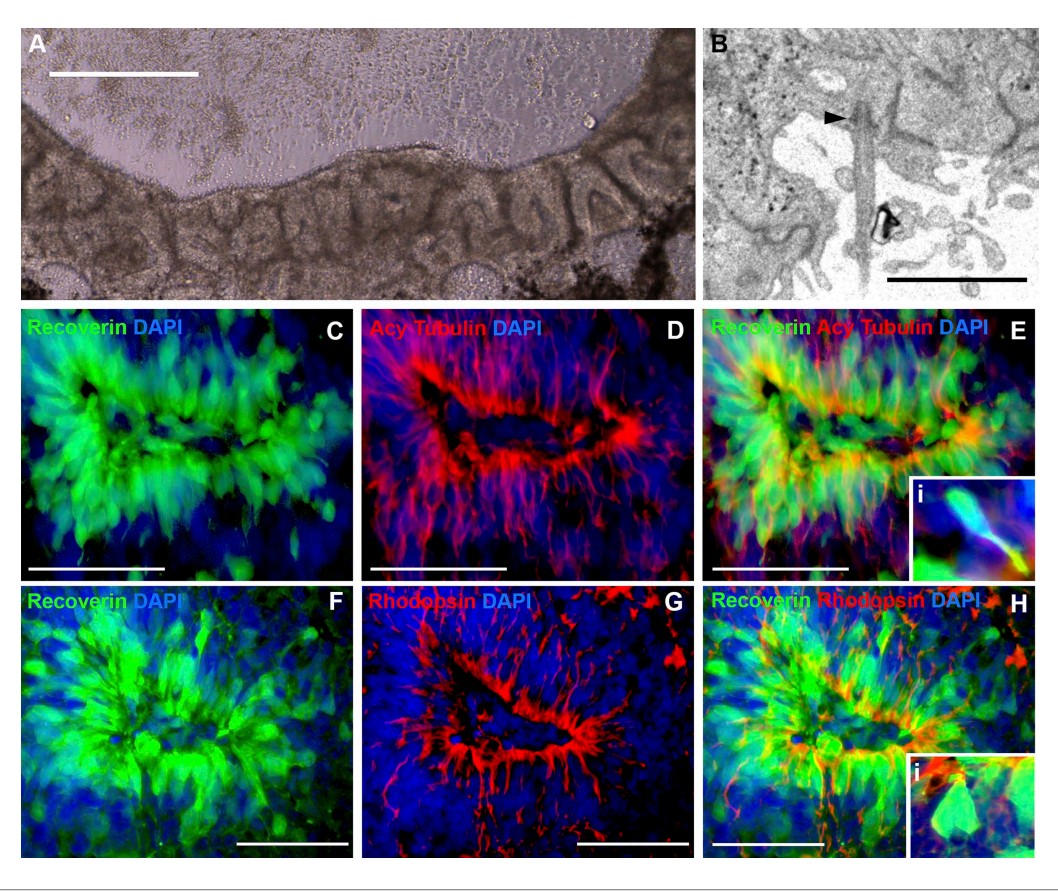

**Figure 4**. iPSC-derived USH2A-associated neural retinal rosettes consist predominantly of rod photoreceptor cells. (**A**) Morphological depiction of the neural retina at 120 days post-differentiation. (**B**) TEM analysis of neural rosettes demonstrates the existence of cilia with clearly identifiable basal bodies. (**C–H**) Immunocytochemical analysis targeted against the rod photoreceptor markers recoverin and rhodopsin (**C–E** and **Ei**-high magnification inlay), and the rod photoreceptor marker recoverin and the connecting cilia marker acetylated tubulin (**F–H** and **Hi**-high magnification inlay).

recoverin positive cells (***Figure 4C,E,F, and H***, and ***Figure 2—figure supplement 1D–F***, green) with acetylated tubulin (***Figure 4D,E***, red) and rhodopsin (***Figure 4G,H***, and ***Figure 2—figure supplement 1D–F***, red) positive structures concentrated at the luminal surface of the rosettes. TEM analysis further confirmed the existence of cilia with clearly identifiable basal bodies (***Figure 4B***, arrowhead).

In an attempt to determine the developmental timeline of retinal gene expression, a series of rt-PCR and western blot analyses were performed on retinal progenitor cultures isolated at 60, 90, and 120 days post-differentiation. The retinal transcripts *PAX6*, *OTX2*, *CRX*, *NRL*, recoverin, and rhodopsin were detected in differentiating patient-specific iPSCs as early as 60 days post-differentiation (***Figure 5A***). However, the mature rod photoreceptor protein recoverin and the rod photopigment rhodopsin were not detected until 90 and 120 days post-differentiation, respectively (***Figure 5B***). Similarly, the cone photopigments blue opsin and red/green opsin were first detected at 90 and 120 days post-differentiation, respectively (***Figure 5B***).

To determine whether *USH2A* was expressed in iPSC-derived neural retina and whether the mutations we identified in the genomic DNA affected the *USH2A* gene product, rt-PCR analysis of RNA isolated from human control retina, control iPSC-derived photoreceptor precursor cells, and patient-specific iPSC-derived photoreceptor precursor cells were performed. As shown in ***Figure 6***, RNA isolated from patient-specific iPSC-derived photoreceptor precursor cells revealed that the suspected splice site mutation identified in intron 40 of the proband caused exonification of the intron (***Figure 6A***). Introduction of intronic sequence between exons 40 and 41 results in a frame shift and introduction of a premature stop codon. Similarly, Sanger sequencing of rt-PCR products generated from the iPSC-derived

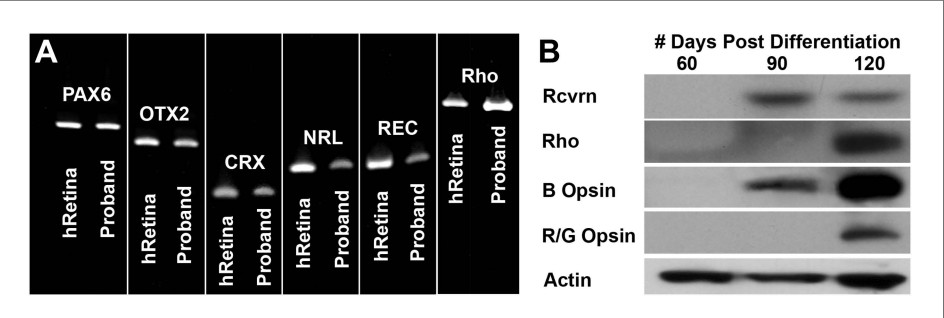

**Figure 5**. Developmental timeline of neural retina marker expression. (**A**) RT-PCR analysis of *USH2A* and human control neural retina for expression of the retinal transcription factors/photoreceptor markers PAX6, OTX2, CRX, NRL, recoverin, and rhodopsin at 60 days post-differentiation. (**B**) Western blot analysis of *USH2A* neural retina for expression of the retinal photoreceptor markers recoverin, rhodopsin, blue cone opsin and red/green cone opsin at 60, 90, and 120 days post-differentiation. Although retinal transcripts can be detected as early as 60 days post-differentiation, mature photoreceptor proteins such as recoverin, rhodopsin, and the cone opsins could not be detected until 90 to 120 days post-differentiation.

photoreceptor precursor cells allowed us to confirm the single point mutation identified within exon 63 of the patients' DNA (*Figure 6B*). In an attempt to determine the pathophysiological mechanism of these mutations, a series of western blot experiments were performed to look for evidence of mutation-induced apoptosis, ubiquitination, and ER stress. As shown in *Figure 6C*, when compared to normal human retina, iPSC-derived photoreceptor precursor cells from a normal control and a separate RP patient with known disease-causing mutations and pathophysiology (MAK associated RP caused by nonsense mediated decay of the transcript), the proband was found to have increased expression of the markers GRP78 and GRP94 indicative of protein misfolding and subsequent ER stress (*Obeng et al., 2006*; *Lind et al., 2013*).

For patients who have lost the majority of their retinal photoreceptor cells and in turn the majority of their vision, our ultimate goal is to develop a patient-specific autologous cell replacement strategy that can repopulate the outer retina with functional photoreceptors. The ability to transplant such cells into animals with retinal degeneration will also be valuable as part of the vision scientist's armamentarium for exploring the pathophysiologic mechanism of specific mutations that are identified in patients. To test whether patient-specific photoreceptor precursor cells isolated from the neural retina layer of human iPSC-derived eyecups could give rise to new photoreceptor cells in animals for the latter purpose, a series of transplantation experiments were performed. It has been previously shown that the optimal cell type for retinal transplantation is the post-mitotic photoreceptor precursor cell (*MacLaren et al., 2006*; *Pearson et al., 2012*). Prior to transplantation, 150-day neural rosettes were dissected free from their surrounding tissues and plated onto fresh tissue culture plates to determine whether these cells would maintain a photoreceptor cell identity following dissociation. To test this, cell cultures were infected 3 days after plating with a lentiviral vector driving expression of GFP under control of the rhodopsin kinase promoter. 2 weeks after plating, post-mitotic rhodopsin kinase positive photoreceptor precursor cells were abundant (*Figure 7A,B*). In many instances, clusters of the rhodopsin kinase positive cells realigned in a polarized photoreceptor cell fashion and extended axon-like projections (*Figure 7B*, arrowhead) and outer-segment-like processes (*Figure 7B*, arrow). Following the confirmation of a stable photoreceptor cell fate, newly generated 150-day photoreceptor precursor cells were transplanted into the subretinal space of P4 immunodeficient *Rag1⁻/⁻* x *Crb1⁻/⁻* mice. *Crb1⁻/⁻* mice were chosen as recipient animals because they exhibit a relatively slow retinal degeneration and also because they are known to be more conducive to cellular integration with proper rod photoreceptor morphology following transplantation than other retinal degenerative strains (*Barber et al., 2013*).

2 weeks after transplantation, extensive cellular integration was detected using the human Tra-1-85 blood antigen as a marker of human cells (*Figure 7C–F*, *Figure 7—figure supplement 1*, Sham injection control, red). The integrated cells expressed the mature photoreceptor marker recoverin, extended axonal projections toward the inner plexiform layer (*Figure 7D–F* arrowheads), and developed

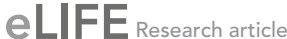

**Figure 6**. Confirmation of genomic *USH2A* variants in iPSC-derived neural retina. (**A**) RT-PCR analysis of USH2A exons 39 to 41 in human control retina (***DePristo et al., 2011***), human control iPSC-derived neural retina (***Baux et al., 2007***), and human RP iPSC-derived neural retina. An intronic splice site mutation in intervening sequence 40 of the USH2A gene results in the introduction of a pseudoexon (IVS40 Red) causing a translation frameshift and a premature stop codon. (**B**) RT-PCR analysis of *USH2A* exon 62 to 63 in human control human retina (***DePristo et al., 2011***), human control iPSC-derived neural retina (***Baux et al., 2007***), and human RP iPSC-derived neural retina. A single heterozygous point mutation identified by whole exome sequencing (Arg4192His) was confirmed in the proband's retinal transcript. (**C**) Western blot analysis of protein isolated from human control retina and iPSC-derived photoreceptor precursor cells obtained from the proband, an unaffected control, and a separate RP patient with known disease pathophysiology for expression of the ER-stress markers GRP78 and GRP94. Elevated expression of both GRP78 and GRP94 suggests that the mutations identified within the proband result in protein misfolding and ER-stress. **p<0.001

inner- and outer-segment-like projections that extended toward the underlying RPE layer (***Figure 7D–F*** arrows).

## Discussion

Patient-specific iPSC-derived retinal cells are a valuable new tool for investigators seeking to understand and treat degenerative retinal diseases. These cells will allow scientists to explore the pathophysiology of human diseases in ways that were previously possible only in animal models. They will also be useful for evaluating potential therapies ranging from high-throughput screens of small molecule drugs to the comparison of gene replacement constructs containing different promoters or packaged in different vectors. For rare autosomal recessive disorders such as *USH2A*-associated RP, there is often little statistical evidence for the pathogenicity of one or both of a patient's putative disease-causing mutations. In such cases, iPSC-derived retinal cells may be useful for confirming the pathogenicity of an unusual genotype before embarking upon an invasive therapy like subretinal administration of viral-mediated gene therapy.



**Figure 7**. *USH2A*-associated photoreceptor precursor cells integrate into the dystrophic mouse retina and develop into mature photoreceptor cells. (**A** and **B**) Microscopic analysis of rhodopsin kinase GFP expression in 150-day photoreceptor precursor cells at 14 days post-plating. (**C**–**E**) Immunocytochemical analysis performed on the retinas of *Rag*⁻/⁻ x *Crb1*⁻/⁻ degenerative eyes 14 days after receiving subretinal injections of patient-specific photoreceptor precursor cells targeted against expression of the human cell antigen Tra-1-85 (**C**–**F**) and the photoreceptor marker recoverin (**D** and **E**). Scale bar = 50 µm.

The following figure supplements are available for figure 7:

**Figure supplement 1**. *USH2A*-associated photoreceptor precursor cells integrate into the dystrophic mouse retina.

In this study, we combined sequencing and iPSC technologies to identify and confirm the pathogenicity of disease-causing mutations in an isolated patient with autosomal recessive RP. Exome sequencing of this individual revealed more than 400 plausible disease-causing sequence variations in more than 300 genes. Two of these were present in genes known to cause autosomal recessive RP (an Arg4192His variant in *USH2A* and a Thr1428Met in *ABCA4*), but a second disease-causing allele could not be found in either of these genes despite Sanger sequencing of the entire coding sequence. We were fortunate that Vaché et al. identified and reported the *USH2A* variant in IVS 40 that proved to be our patient's second disease-causing allele. Non-exomic mutations are not an uncommon cause of recessive diseases. In some instances (e.g., *CEP290*-associated LCA) they account for the largest proportion of disease-causing alleles (***Stone, 2007***). As the iPSC technology becomes more routine, the evaluation of RNA from iPSC-derived retina may become a common step in the analysis of exome sequencing data when two clearly disease-causing alleles cannot be identified in known disease genes. To confirm the pathogenicity of the *USH2A* variants we observed in our patient, *USH2A* transcripts were analyzed using RNA isolated from iPSC-derived photoreceptor precursor cells obtained from the proband, an unaffected control and human donor retina. The suspected splice site mutation within IVS40 was shown to cause exonification of the intron, a translation frameshift and a premature stop codon. The transcription of the patient's *USH2A* missense mutation was confirmed in a similar fashion. In combination, these mutations

resulted in an upregulation of the markers GRP78 and GRP94 indicative of protein misfolding and subsequent ER stress.

Following transplantation into neonatal retinal degenerative *Crb1* mutant mice, our patient's photoreceptor precursor cells integrated into the outer nuclear layer and differentiated into morphologically and immunohistochemically recognizable photoreceptors. This finding is compatible with our patient's history of normal vision until the third decade of life; that is, his *USH2A* mutations do not appear to cause a gross developmental abnormality of photoreceptor cells. It is also noteworthy that transplantable photoreceptor precursor cells could be generated from the skin of a patient in the seventh decade of life. The transplantation of photoreceptor precursor cells derived from early postnatal retina or embryonic stem cells has been accomplished before by other groups (*MacLaren et al., 2006*; *La Torre et al., 2012*; *Pearson et al., 2012*), but to our knowledge, this is the first time that photoreceptor precursor cells have been derived from an adult human RP patient and successfully transplanted into mice. However, much work remains to be done before such transplants could be considered evidence for the feasibility of sight-restoring iPSC-derived treatments in humans. For example, synaptic connectivity will need to be demonstrated ultrastructurally and electrophysiologically, and useful vision in treated animals will need to be demonstrated with an array of psychophysical approaches. Longevity of the transplanted cells will be important to demonstrate as will the cells' lack of tumorigenicity. For such transplants to have practical clinical utility it will also be important to show that the transplanted cells can integrate and function in the retinas of animals with advanced stages of many different molecular types of retinal degeneration.

Unlike many of the genes targeted in current clinical trials, *USH2A* is very large and will be impossible to package into the viral vectors in current clinical use. The coding sequence of *USH2A* is >18 kb, and the packaging limits of AAV and EIAV vectors are about 3 kb and 10 kb, respectively. For genes such as *USH2A*, vectors with large carrying capacities such as HSV1, which can accommodate inserts of up to 40 kb in size (*Thomas et al., 2003*), may be useful. In addition to the vector limitations, the proper stoichiometry of the delivered message will also be critical for some genes. It is possible that patients like the one reported here, with a truncated protein encoded by one allele and a misfolded protein encoded by the other, may require a different promoter and multiplicity of infection than a different patient who harbors two stop mutations in *USH2A*. Patient-specific iPSC-derived retinal cells may be useful in choosing the optimal gene therapy strategy for each individual.

For late-stage *USH2A*-associated RP where significant photoreceptor cell loss has occurred, cell replacement strategies will be required to restore vision. In some cases, correcting the genetic defect responsible for the photoreceptor degeneration will be necessary before transplanting photoreceptor cells into the patient. The development of site specific TALEN- or CRISPR-based genome editing approaches, which allow for the targeted correction of patient-specific cells in vitro, is proving to have great utility for endogenous gene correction (*Hockemeyer et al., 2011*; *Ding et al., 2013*; *Wang et al., 2013*). For instance, Hockemeyer et al. recently demonstrated that for the five genomic sites targeted, pluripotent iPSC clones carrying transgenes solely at TALEN-specified loci could be obtained (*Hockemeyer et al., 2011*). Similarly using the CRISPR/CAS-9 system, Wang et al. recently demonstrated the ability to simultaneously manipulate five separate genetic loci in mouse ES cells (*Wang et al., 2013*). A major advantage of these approaches is that unlike exogenous gene addition in which promoter strength and multiplicity of infection may require patient-specific adjustment, TALEN- and CRISPR-mediated correction have the advantage that the gene remains under the control of the endogenous promoter. For recessive diseases such as *USH2A*-associated RP, correction of a single disease-causing allele should be sufficient for a therapeutic effect. One could test this directly by transplanting corrected cells into retinal degenerative mice and determining whether cells with both alleles corrected behave differently than those with one or neither allele corrected.

It is possible that for some individuals with late-onset disease, such as the patient described in this report, genetic correction prior to transplantation may not be required. That is, if the patient's native photoreceptor cells develop and function normally until the third decade of life, as suggested by our current data and the patient's clinical history, it is possible that replacement of lost photoreceptors with iPSCs that have not been genetically modified could be a reasonably durable treatment.

An alternative approach to patient-specific autologous cell replacement that would also not require genetic manipulation of the donor cell population prior to delivery would be the use of either genetically unmatched ES or tissue-specific precursor cells. This approach is currently in clinical trial for the treatment of Stargardt macular dystrophy and age-related macular degeneration (AMD) (*Schwartz et al., 2012*).

In these studies, ES cell-derived retinal-pigmented epithelial cells are delivered to immune suppressed patients as subretinal bolus cell injections (*Schwartz et al., 2012*). However, there are considerable disadvantages to using allogeneic cells. Retinas injured by inherited disease often have a loss of integrity of the blood retinal barrier, which would allow the patient's peripheral immune system access to the transplanted cells. This type of transplant approach would likely require life-long immunomodulation, (*Hambright et al., 2012*) putting patients at further risk.

In summary, by combining next-generation and Sanger sequencing with iPSC technologies we were able to demonstrate the pathogenicity of two disease-causing mutations in a patient with non-syndromic *USH2A*-associated RP. We also demonstrated that keratinocytes cultured from a patient in the seventh decade of life can be reprogrammed into iPSCs and differentiated into a multi-layered eyecup-like structure with immunohistochemical and ultrastructural features of human retinal precursor cells. Finally, we showed that these patient-derived retinal precursor cells have the ability to integrate into the developing mouse retina and to form morphologically and immunohistochemically recognizable photo-receptor cells. These findings will enable patient-specific studies of disease mechanism, gene correction, and photoreceptor cell transplantation for many different types of human retinal degeneration.

## Materials and methods

### Ethics statement

All experiments were conducted with the approval of the University of Iowa Animal Care and Use Committee (Animal welfare assurance #1009184) and the University of Iowa Internal Review Board (IRB # 200202022). All experiments were consistent with the ARVO Statement for the Use of Animals in Ophthalmic and Vision Research and the Treaty of Helsinki.

### Patient-derived cells

After informed consent, skin biopsies were collected from a 62-year-old patient with an unknown cause of RP, three patients with known causes of retinal disease and three individuals without eye disease, and these were used for fibroblast and keratinocyte isolation (as described previously [*Bickenbach, 2005*; *Tucker et al., 2013*]). Cells were expanded and targeted for iPSC generation.

### iPSC generation

iPSCs were generated from human patient-specific keratinocytes via infection with four separate non-integrating Sendai viruses, each of which were designed to drive expression of one of four transcription factors: OCT4, SOX2, KLF4, and c-MYC (A1378001, Invitrogen, Grand Island, NY). Keratinocytes plated on six-well tissue culture plates were infected at an MOI of 5. At 12–16 hr post-infection, cells were washed and fed with fresh growth media (Epilife media with keratinocyte supplement [Invitrogen] and 0.2% primocin [Invivogen]). At 7 days post-infection, cells were passaged onto six-well Synthemax cell culture dishes at a density of 300,000 cells/well and fed every day with pluripotency media (DMEM F-12 media [Gibco], 20% knockout serum replacement [Gibco], 0.0008% beta-mercaptoethanol [Sigma-Aldrich, St. Louis, MO], 1% 100 × NEAA [Gibco], 100 ng/ml bFGF [human] [R&D], and 0.2% primocin [Invivogen]. At 3 weeks post-viral transduction, iPSC colonies were picked, passaged, and clonally expanded on fresh Synthemax plates. During reprogramming and maintenance of pluripotency, cells were cultured at 5% $CO_2$, 5% $O_2$, and 37°C.

### iPS cell differentiation

To maintain pluripotency, adult-derived iPSCs were cultured in xeno/feeder free cell culture media. To initiate differentiation, iPSCs were removed from the culture substrate via manual passage using Stem Passage manual passage rollers (Invitrogen), resuspended in embryoid body (EB) media (DMEM F-12 media [Gibco] containing 10% knockout serum replacement [Gibco], 2% B27 supplement [Gibco], 1% N2 supplement [Gibco], 1% L-glutamine [Gibco], 1% 100 × NEAA [Gibco], 0.2% primocin [Invivogen], 1 ng/ml noggin [R&D Systems, Minneapolis, MN], 1 ng/ml Dkk-1 [R&D Systems], 1 ng/ml IGF-1 [R&D Systems], and 0.5 ng/ml bFGF [R&D Systems]), and plated at a density of ~50 cell clusters/cm² on ultra low adhesion culture plates (Corning, Lowell, MA). Cell clusters were cultured for 5 days as indicated above, after which the EBs were removed, washed, and plated at a density of 25–30 EBs/cm² in fresh differentiation media 1 (DMEM F-12 media [Gibco], 2% B27 supplement [Gibco] 1% N2 supplement [Gibco], 1% L-glutamine [Gibco], 1% 100 × NEAA [Gibco] 10 ng/ml noggin [R&D Systems], 10 ng/ml Dkk-1 [R&D Systems], 10 ng/ml IGF-1 [R&D Systems] and 1 ng/ml bFGF [R&D Systems]) in six-well

Synthemax culture plates. Cultures were fed every other day for 10 days with differentiation media 1. For the following 6 days, cultures were fed with differentiation media 2 (differentiation media 1 + 10 μM of the Notch signaling inhibitor, DAPT [Calbiochem, Gibbstown, NJ]). For the following 12 days, cultures were fed with differentiation media 3 (differentiation media 2 + 2 ng/ml of aFGF [R&D Systems]). To enhance pigmentation and eyecup-like structure formation, cells were passaged at day 50–70 in fresh cell culture plates coated as described above and cultured for up to 150 days in differentiation media 4 (DMEM F-12 media [Gibco], 2% B27 supplement [Gibco] 1% N2 supplement [Gibco], 1% L-glutamine [Gibco], 1% 100 × NEAA [Gibco]).

## Teratoma formation

To validate that generated iPSCs were pluripotent, teratomas were generated by IM injection of $2.5 \times 10^6$ undifferentiated iPSCs into immunodeficient (SCID) mice. After 90 days, tumors were excised, fixed, paraffin embedded, and sectioned.

## Histology

Teratomas were fixed in 10% formalin for 24 hr prior to dehydration and mounting in paraffin wax (VWR). Samples were sectioned at 6 μm and H&E staining was performed using standard protocols.

## Immunostaining

Cells were fixed in a 4% paraformaldehyde solution and immunostained as described previously (*Tucker et al., 2010*, *2011*, *2013*). Briefly, cells/tissues were incubated overnight at 4°C with antibodies targeted against either GFAP (MAB360; Millipore, Billerica, MA) or αSMA (ab5694; Abcam, Cambridge, MA) for teratoma formation or ZO-1 (MABT11; Millipore), PAX6 (MAB5552; Millipore), bestrophin 1 (AB14929; Abcam), RPE65 (MAB5428; Millipore), acetylated tubulin (t7451; Sigma Aldrich), recoverin (AB5585; Millipore), and rhodopsin (MAB5316; Millipore) for retinal differentiation. Subsequently, Cy2- or Cy3-conjugated secondary antibodies were used (Jackson Immunochem, West Grove, PA), and the samples were analyzed using confocal microscopy. Microscopic analysis was performed such that exposure time, gain, and depth of field remained constant between experimental conditions.

## Transmission electron microscopy (TEM)

Eyecups at 150 days post-differentiation were fixed in one half strength Karnovsky fixative as described previously (PMID 17591911), followed by osmication, dehydration, and embedment in Epon resin. All procedures took place in the tissue culture dish. After polymerization, blocks were removed and trimmed, ultrathin sections were collected on formvar coated grids, and samples were imaged on a JEOL JEM1230 transmission electron microscope.

## Immunoblotting

For Western blot analysis, undifferentiated and differentiated iPSCs were homogenized in lysis buffer (50 mM Tris-HCl, pH 7.6, 150 mM NaCl, 10 mM $CaCl_2$, 1% triton X-100, 0.02% $NaN_3$ [Sigma-Aldrich]) and centrifuged. Supernatants were isolated and protein concentrations determined using a BCA protein assay (Pierce Chemicals, Rockford, IL). Equivalent amounts of protein (50 μg) were subjected to SDS-PAGE (8–10% acrylamide), transferred to PVDF, and probed with primary antibodies targeted against recoverin (AB 5585; EMD Millipore), rhodopsin (MAB5316; EMD Millipore), blue opsin (AB5407; EMD Millipore), red/green opsin (AB5405; EMD Millipore), GRP 78 (SC-376768; Santa Cruz), GRP 94 (SC-53929; Santa Cruz) and Actin (AB20272; Abcam; used as a loading control). Blots were visualized with ECL reagents (GE healthcare, Piscataway, NJ) and exposed to X-ray film (Fisher, Pittsburg, PA).

## RNA isolation and rt-PCR

Total RNA was extracted using the RNeasy Mini-kit (Qiagen, Valencia, CA) following the provided instructions. Briefly, cells were lysed, homogenized, and ethanol was added to adjust binding conditions. Samples were spun using RNeasy spin columns, washed, and RNA was eluted using RNase-free water. 1 μg of RNA was reverse transcribed into cDNA using the random hexamer (Invitrogen, Carlsbad, CA) priming method and Omniscript reverse transcriptase (Qiagen). All PCR reactions were performed in a 40 μl reaction containing 1× PCR buffer, 1.5 mM $MgCl_2$, 0.2 mM dNTPs, 100 ng of DNA, 1.0 U of AmpliTaq Gold (Applied Biosystems, Foster City, CA) and 20 pmol of each gene-specific primer. All cycling profiles incorporated an initial denaturation temperature of 94°C for 10 min followed by 35 amplification cycles with the following conditions, 30 s at 94°C, 30 s at annealing temperature of

each primer, and 1 min at 72°C with a final extension at 72°C for 10 min. PCR products were separated by electrophoresis on 2% agarose gels (Invitrogen). Gene-specific primers (Invitrogen) are given in *Supplementary file 1A*.

## DNA extraction

Blood samples were obtained from all subjects. DNA was extracted by following the manufacturers specifications for whole blood DNA extraction using Gentra Systems' Autopure LS instrument.

## Exon capture

Targeted enrichment of exons was performed using the Agilent SureSelect All Exon Capture platform per manufacturer's instructions. This capture platform includes 38 Mb of targeted features.

## Next-generation DNA sequencing

Sequencing of the captured genomic DNA was performed following the manufacturer's instructions on an Illumina HiSeq sequencer at the Hudson Alpha Institute in Huntsville, Alabama.

## Automated DNA sequencing

Variants detected by next-generation sequencing were confirmed using automated Sanger sequencing using dye termination chemistry on an ABI 3730 sequencer. All sequencing was bi-directional.

## Transplantation

Newly generated 150-day photoreceptor precursor cells were transplanted into the subretinal space of P4 immune compromised *Rag1*$^{-/-}$ x *Crb1*$^{-/-}$ mice. 2 weeks after transplantation, mice were sacrificed and eyes were removed for histological analysis, fixed in 4% paraformaldehyde for 12 hr, embedded and sectioned on a cryostat. Sectioned mouse eyes were used for immunofluorescence analysis. Sections were blocked in 10% goat serum 3% BSA and 0.2% triton X-100 for 1 hr, and then incubated with anti-Tra-1-85 (MAB4385; Millipore), and recoverin antibodies (AB5585; Millipore) followed by visualization with Cy2- and Cy3-conjugated secondary antibodies (Jackson Immunochem, West Grove, PA). Sections were assessed by confocal microscopy.

## Additional information

### Funding

| Funder | Grant reference number | Author |
| --- | --- | --- |
| Howard Hughes Medical Institute | | Edwin M Stone |
| NIH Directors New Innovator Award | 1-DP2-OD007483-01 | Budd A Tucker |
| National Eye Institute | EY017451 | Robert F Mullins, Edwin M Stone |
| Foundation Fighting Blindness | | Budd A Tucker, Edwin M Stone |
| Stephen A Wynn Foundation | | Budd A Tucker, Robert F Mullins, Edwin M Stone |
| Grousbeck Family Foundation | | Budd A Tucker, Robert F Mullins, Edwin M Stone |
| Leo, Jacques and Marion Hauser Family Vision Restoration Fund | | Budd A Tucker, Robert F Mullins, Edwin M Stone |

The funders had no role in study design, data collection and interpretation, or the decision to submit the work for publication.

### Author contributions

BAT, RFM, EMS, Conception and design, Acquisition of data, Analysis and interpretation of data, Drafting or revising the article, Contributed unpublished essential data or reagents; LMS, Acquisition of data, Contributed unpublished essential data or reagents; KA, MEE, EK, MJR, AVD, TAB, Acquisition of data, Drafting or revising the article

## Ethics

Human subjects: All experiments were conducted with the approval of the University of Iowa Internal Review Board (IRB # 200202022). All subjects provided written informed consent for this research study, which was approved by the Institutional Review Boards of the participating centers and adhered to the tenets set forth in the Declaration of Helsinki.

Animal experimentation: All experiments were conducted with the approval of the University of Iowa Animal Care and Use Committee (Animal welfare assurance #1009184).

## Additional files

### Supplementary files

• Supplementary file 1. (**A**) Prioritization of exome variants. (**B**) Gene-specific primer sequences used for rt-PCR. F = forward primer and R= reverse primer.

### Major datasets

The following previously published datasets were used:

| Author(s) | Year | Dataset title | Dataset ID and/or URL | Database, license, and accessibility information |
| --- | --- | --- | --- | --- |
| The 1000 Genomes Project Consortium | 2012 | Sequence Reads—All Exomes | ftp://ftp-trace.ncbi.nih.gov/1000genomes/ftp/data/ | Instructions on accessing data at the 1000 Genomes website: http://www.1000genomes.org/data#DataAccess. |
| NHLBI GO Exome Sequencing Project (ESP) | 2012 | Exome Variant Server | ESP-6500SI; http://evs.gs.washington.edu/EVS/ (Version 2) | Used release dated October 31, 2012; current release available from http://evs.gs.washington.edu/EVS/. |

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
