## [Decision Letter]

Thank you for sending your work entitled “Patient-specific iPSC-derived photoreceptor precursor cells as a means to investigate and treat retinitis pigmentosa” for consideration at *eLife*. Your article has been favorably evaluated by a Senior editor and 3 reviewers, one of whom is a member of our Board of Reviewing Editors.

The Reviewing editor and the other reviewers discussed their comments before we reached this decision, and the Reviewing editor has assembled the following comments to help you prepare a revised submission.

Tucker et al. describe a series of experiments in which a 62-year-old patient with autosomal recessive RP had his DNA analyzed by exome sequencing. Presumptive disease-causing mutations were defined in the USH2A gene and the patient’s keratinocytes were harvested and converted to iPSC cells. The iPSCs were differentiated to form retina-like eye-cup structures in culture (including highly differentiated RPE), the RNA splicing defect associated with one of the patient’s disease alleles was demonstrated in iPSC-derived photoreceptor-like cells in culture, and the iPSCs were transplanted into a recipient mouse retina where they integrate into the photoreceptor layer.

The significance of this work is that it demonstrates current state-of-the-art technologies both with respect to disease mutation identification and, more importantly, with respect to the reprogramming of human keratinocytes into ocular cell types. The study is one of the first to apply iPSC technology to elucidate pathogenetic mechanisms (i.e., the RNA splicing defect). The authors’ success with obtaining iPSC cells from an older human subject is especially noteworthy. This manuscript will be of broad interest to those working on iPSC-based therapies. The details of the iPSC methodology will be especially useful to the ocular therapeutics community.

The experiments purporting to show differentiation of transplanted human photoreceptors are quite modest and should be described as such. There is a large scope here for artifacts. Unambiguous markers and clear morphology are required, and transplantation into different animal models may also be required. This is a central issue because at present no one has convincingly demonstrated successful transplantation of human photoreceptors. Thus, it would be useful to see a more detailed characterization of the transplanted human photoreceptors in the *Rag* KO/*Crb1* KO mouse retina. For example, a transmission EM analysis of transplanted human photoreceptors would address, at the ultra-structural level, the extent to which the human cells have acquired the morphological hallmark photoreceptors, including outer segment production. In the best case, this would include an EM label for human vs. mouse cells – immuno-EM, presumably. A second approach to this question would be to immunostain for Tra-1-85 together with antibodies for outer segment, connecting cilium and presynaptic nerve terminal components. In the Abstract and Discussion, the authors refer to the transplanted photoreceptors as “morphologically normal rod photoreceptors”, but absent a more detailed characterization that is over-stating the data. We also note that a complete analysis would include functional assays, such as single cell recording of the light response. We appreciate that this is beyond the scope of the present study, but it would be useful to summarize in the Discussion the work that remains to be done.

What follows are a series of specific comments related to the iPSC differentiation and transplantation experiments:

1) The rationale for transplanting the iPSC cells to determine if cell replacement therapy would be possible is not clearly described, given that the mutations are still present in the new photoreceptors and the effects this may have on cell survival and function are unknown and were not investigated.

2) The authors show a single example of an eye cup-like structure in Figure 2. The neural retinal region of the cell clump is made up of individual neural rosettes as opposed to the laminated 3-dimensional retinal structures described as eye cup-like by others. Can the authors elaborate on their relatively flattened arrangement? In addition, how often were cell clumps with this organization observed within the cultures? The authors state that “multiple eye-cups, each at different developmental stages, were present within the same well”. Can they elaborate on the different stages of retinal development observed and explain why this may be the case given their stepwise differentiation protocol?

3) Can the authors clarify what they mean by “presumptive outer segments”? Acetylated tubulin is a marker of microtubules such as those present in the connecting cilium but it does not label photoreceptor outer segments. Immunostaining for acetylated tubulin in the retina should result in punctuate staining at the site of the connecting cilium, but not throughout the rest of the photoreceptor.

4) The resolution of the TEM image shown in Figure 4 (to corroborate the presence of the connecting cilia) is too poor to clearly see the presence of microtubules and there is no demonstration of the photoreceptor specific 9+0 arrangement. It is therefore unclear if this is in fact a sensory non-motile cilium. Can the authors provide further TEM images to verify the claims made regarding the presence of photoreceptor connecting cilia?

5) Given that the authors have not demonstrated clear labeling or ultrastructural morphology for either inner or outer segments present in the cultured iPS cell-derived photoreceptors, it is difficult to conclude that the rhodopsin staining observed in Figure 4 is in the “presumptive outer segment”. Can the authors provide further data to support this claim?

6) The authors conclude “our patient’s photoreceptor precursor cells integrated into the outer nuclear layer and differentiated into morphologically normal rod photoreceptors”. This statement implies complete photoreceptor morphology in iPS cell-derived integrated photoreceptors, including synaptic boutons, cell processes, cell body, and outer segment formation.

7) Is two weeks sufficient time to observe mature morphological features in integrated human iPS cell-derived photoreceptors? Can the authors comment on the time taken for the morphological maturation of human photoreceptors, especially as they state that “retinal differentiation of human cells takes significantly longer” than mouse cells?

8) Given the non-specific secondary staining of the retinal vessels in Figure 7, can the authors provide an image of the injection site from the primary omitted control staining, as it is difficult to determine the positive staining from the non-specific background staining when using antibodies raised in mouse on mouse tissue. Other examples of human specific markers, ideally neural or photoreceptor specific, to clearly identify the iPS cell-derived cells from the host murine photoreceptors would be useful.

9) Figure 6: are those individual bands from the same blots/same exposure? Are their intensities directly comparable? Running the samples side-by-side on the same gel and same blot is the cleanest way to do this.

---

## [Author Response]

*The experiments purporting to show differentiation of transplanted human photoreceptors are quite modest and should be described as such. There is a huge scope here for artifacts. Unambiguous markers and clear morphology are required, and transplantation into different animal models may also be required. This is a central issue because at present no one has convincingly demonstrated successful transplantation of human photoreceptors. Thus, it would be useful to see a more detailed characterization of the transplanted human photoreceptors in the Rag KO/Crb1 KO mouse retina. For example, a transmission EM analysis of transplanted human photoreceptors would address, at the ultra-structural level, the extent to which the human cells have acquired the morphological hallmark photoreceptors, including outer segment production. In the best case, this would include an EM label for human vs. mouse cells – immuno-EM, presumably. A second approach to this question would be to immunostain for TRA1-85 together with antibodies for outer segment, connecting cilium, and presynaptic nerve terminal components*.

We agree that if the primary purpose of the paper were to convincingly demonstrate the feasibility of a stem-cell-based photoreceptor replacement therapy, a complete demonstration of photoreceptor cell identity, synaptic connectivity, and electrophysiologic and psychophysical function would be required. In fact, such studies are underway in our laboratories using a variety of immune deficient retinal degenerative mouse strains (Rag-Rho^-/-^, Rag-Mak^-/-^, Rag-Crb1^-/-^, Rag-Prph2^-/-^ & Rag-Cep290^-/-^) and a variety of different patient specific cell lines (from control individuals and patients affected with heritable retinal degeneration). However, the primary goal of the current study was to demonstrate that photoreceptor precursor cells could be generated from the skin of a patient in the seventh decade of life affected with a rare inherited retinal disease (retinitis pigmentosa), and that these cells could be used to demonstrate the pathogenicity of a recently described non-exomic mutation in a large retina-specific gene (*USH2A*). Given this, we have decided to remove the words “and treatment” from the title, because we now realize that these words would suggest to some readers that the types of transplantation detail asked for by the reviewers would be found in this paper.

The rationale for the transplantation portion of this paper was to show that the identified *USH2A* mutations do not result in gross developmental abnormalities of the photoreceptors; that is, it is a component of our investigation of the pathophysiology of the mutations. However, in response to the reviewers’ comments we have provided further information regarding secondary control staining (Figure 7—figure supplement 3 shows Tra-1-85 and Cy3 secondary staining of the contralateral sham injected eye).

*In the Abstract and Discussion, the authors refer to the transplanted photoreceptors as “morphologically normal rod photoreceptors”, but absent a more detailed characterization that is over-stating the data*.

As noted above, our intent was to demonstrate that the patient’s *USH2A* mutations did not cause gross developmental abnormalities of the photoreceptor structure. However, we agree with the reviewers that without a more complete characterization, the phrase “morphologically normal rod photoreceptors” is an overstatement. Therefore, we have altered the text to say “morphologically and immunohistochemically recognizable photoreceptors” instead of “morphologically normal rod photoreceptors”.

*We also note that a complete analysis would include functional assays, such as single cell recording of the light response. We appreciate that this is beyond the scope of the present study, but it would be useful to summarize in the Discussion the work that remains to be done*.

We agree with the reviewers and we have added a paragraph to the Discussion to outline some of the work that will need to be done before asserting the feasibility of iPSC-derived retinal therapies in humans (starting “However, much work remains to be done…”).

*What follows are a series of specific comments related to the iPSC differentiation and transplantation experiments*:

*1) The rationale for transplanting the iPSC cells to determine if cell replacement therapy would be possible is not clearly described, given that the mutations are still present in the new photoreceptors and the effects this may have on cell survival and function are unknown and were not investigated*.

We have added and altered the following sentences in the Results section to clarify our rationale for the transplantation portion of the paper:

“The ability to transplant such cells into animals with retinal degeneration will also be valuable as part of the vision scientist’s armamentarium for exploring the pathophysiologic mechanism of specific mutations that are identified in patients. To test whether patient-specific photoreceptor precursor cells isolated from the neural retina layer of human iPSC-derived eyecups could give rise to new photoreceptor cells in animals for the latter purpose, a series of transplantation experiments were performed.”

*2) The authors show a single example of an eye cup-like structure in*
Figure 2*. The neural retinal region of the cell clump is made up of individual neural rosettes as opposed to the laminated 3-dimensional retinal structures described as eye cup-like by others. Can the authors elaborate on their relatively flattened arrangement? In addition, how often were cell clumps with this organization observed within the cultures? The authors state that “multiple eye-cups, each at different developmental stages, were present within the same well”. Can they elaborate on the different stages of retinal development observed and explain why this may be the case given their stepwise differentiation protocol*?

We use the term “eye-cup-like structures to refer to the circular and bilaminar arrangement of cells with photoreceptor precursors on the inside and RPE precursors on the outside. These eye-cup-like structures are seen routinely in our culture system, i.e., on average there are 2–3 of these structures in each well of a 6 well plate. In some cases, the structures are not as well organized as the one studied most extensively in this paper and consist of RPE precursors surrounding a confluent disk of photoreceptor rosettes. In all cases, the pigmented RPE cells appear first and the nonpigmented photoreceptor precursors appear later, suggesting that the former induce or otherwise give rise to the latter through some as yet uncharacterized mechanism.

To acknowledge and show the variability of the eyecup anatomy we observe, we now include the following sentence in the Results section and include some more images of multiple eyecup-like structures in Figure 2—figure supplement 1:

“In some cases these RPE/neural units extended into C-shaped eyecup-like structures (Figure 2), while in others the central cavity did not form and the structure consisted of a circular sheet of neural cells surrounded by a pigmented epithelium (Figure 2—figure supplement 1).”

The reason that our eyecup like structures develop in a planar fashion is that we grow them in a tissue culture dish and not in suspension culture as reported by Meyer and Eiraku. We have now added the following sentences to explain this, as well as to give a couple of practical advantages of our approach:

“The reason that these eyecup-like structures develop in a planar fashion instead of the spheres reported by both Meyer et al. 2011 and Eiraku et al. 2012 is that we grow these cells under adherent conditions in tissue culture dishes coated with collagen, laminin, and fibronectin, while spheres are grown in suspension. Although there will undoubtedly be advantages of both approaches in different situations, two advantages of this planar system is that the neural cells in the center of the eyecup do not necrose for lack of oxygen and the firm attachment to the underlying substrate allows homogeneous biopsies to be taken from the eyecups for subculture and analysis.”

*3) Can the authors clarify what they mean by “presumptive outer segments”? Acetylated tubulin is a marker of microtubules such as those present in the connecting cilium but it does not label photoreceptor outer segments. Immunostaining for acetylated tubulin in the retina should result in punctuate staining at the site of the connecting cilium, but not throughout the rest of the photoreceptor*.

We agree with the reviewers that antibodies to acetylated tubulin label the connecting cilia of photoreceptors and not the outer segments themselves. In the original manuscript, we used the term “presumptive outer segments” to refer to the tips of the ciliated structures that we observed to express rhodopsin. However, we agree that we need to be very careful not to equate “ciliated structures” with “outer segments”. We have therefore reworded the corresponding section (starting “To determine whether cells contained within non-pigmented neural rosettes”) to reflect this.

*4) The resolution of the TEM image shown in*
Figure 4
*(to corroborate the presence of the connecting cilia) is too poor to clearly see the presence of microtubules and there is no demonstration of the photoreceptor specific 9+0 arrangement. It is therefore unclear if this is in fact a sensory non-motile cilium. Can the authors provide further TEM images to verify the claims made regarding the presence of photoreceptor connecting cilia*?

We agree with the reviewers that Figure 4 does not convincingly show an “axoneme” and we do not have any more convincing TEM images of the cilia of the photoreceptors in hand. We have therefore altered this sentence to read:

“TEM analysis further confirmed the existence of cilia with clearly identifiable basal bodies (Figure 4 arrowhead).”

*5) Given that the authors have not demonstrated clear labeling or ultrastructural morphology for either inner or outer segments present in the cultured iPS cell-derived photoreceptors, it is difficult to conclude that the rhodopsin staining observed in*
Figure 4
*is in the “presumptive outer segment”. Can the authors provide further data to support this claim*?

The term “presumptive outer segment” has been replaced by “acetylated tubulin positive structures concentrated at the luminal surface of the rosettes”.

*6) The authors conclude “our patient's photoreceptor precursor cells integrated into the outer nuclear layer and differentiated into morphologically normal rod photoreceptors”. This statement implies complete photoreceptor morphology in iPS cell-derived integrated photoreceptors, including synaptic boutons, cell processes, cell body, and outer segment formation*.

*USH2A* is known to be a large matrix molecule normally expressed at the base of the connecting cilium and our goal was to show that disease-causing mutations in *USH2A* did not result in gross developmental abnormalities in the inner and outer segments of photoreceptors derived from iPS cells. As noted above, we agree that without a more complete characterization the phrase “morphologically normal rod photoreceptors” is an overstatement. Therefore, we have altered the text in four places to say “morphologically and immunohistochemically recognizable photoreceptor cells”.

*7) Is two weeks sufficient time to observe mature morphological features in integrated human iPS cell-derived photoreceptors? Can the authors comment on the time taken for the morphological maturation of human photoreceptors, especially as they state that “retinal differentiation of human cells takes significantly longer” than mouse cells*?

The reviewers have touched upon a very important point, i.e., the importance of transplanting mature photoreceptor precursor cells rather than immature retinal progenitor cells that require extensive post-transplant differentiation. Following a 2006 publication in *Nature* from Robin Ali’s group describing the transplantation of mature post-mitotic photoreceptor precursor cells, it became clear that for optimal results, the majority of the transplanted cells’ differentiation needs to occur *ex vivo* prior to transplantation. In the experiment reported here, the cells have been differentiated for more than 150 days before the transplantation so that the difference in differentiation time between mouse and human has already been accommodated.

The second aspect of the retinal development question that one needs to consider in a transplantation experiment is how long it takes for the cells to migrate into the proper position, develop polarity, extend axons, establish synapses, etc. The length of time required for human cells to do this in a mouse is unknown at present and was not addressed in this paper. To date, we have only looked at a single time point (2 weeks) but in this one experiment we were pleased to find the degree of integration, polarization, and process extension that we did (which is why we included it in this paper). However, many more experiments with many different time points will be needed to establish the true time course of the post transplant maturation and the physiological function of the transplanted cells.

*8) Given the non-specific secondary staining of the retinal vessels in*
Figure 7*, can the authors provide an image of the injection site from the primary omitted control staining, as it is difficult to determine the positive staining from the non-specific background staining when using antibodies raised in mouse on mouse tissue. Other examples of human specific markers, ideally neural or photoreceptor specific, to clearly identify the iPS cell-derived cells from the host murine photoreceptors would be useful*.

An example of a control sham injection stained with Tra-1-85 and labeled with mouse Cy3 secondary is now presented in Figure 7—figure supplement 3.

*9)*
Figure 6*: are those individual bands from the same blots/same exposure? Are their intensities directly comparable? Running the samples side-by-side on the same gel and same blot is the cleanest way to do this*.

The bands originally presented were from the same blots with the same exposure and were directly comparable. To prevent bleed through and cross contamination between lanes we routinely skip lanes between samples on western blots. Digital removal of the skipped lanes (indicated by the white lines on the originally submitted figure) allowed us to present the data in a more compact fashion. However, to make it clear that the lanes are truly comparable, we have repeated each of the blots in question without skipping lanes and now present them without separation markers.